# The m6A-Related Long Noncoding RNA Signature Predicts Prognosis and Indicates Tumor Immune Infiltration in Ovarian Cancer

**DOI:** 10.3390/cancers14164056

**Published:** 2022-08-22

**Authors:** Rui Geng, Tian Chen, Zihang Zhong, Senmiao Ni, Jianling Bai, Jinhui Liu

**Affiliations:** 1Department of Biostatistics, School of Public Heath, Nanjing Medical University, 101 Longmian Avenue, Jiangning District, Nanjing 211166, China; 2Department of Gynecology, The First Affiliated Hospital of Nanjing Medical University, Nanjing 210029, China

**Keywords:** ovarian serous cystadenocarcinoma, N6-methyladenosine, long noncoding RNAs, prognosis, tumor microenvironment

## Abstract

**Simple Summary:**

OV is the most lethal gynecological malignancy. M6A and lncRNAs have a great impact on OV development and patient immunotherapy response. This study provided an accurate prognostic signature for patients with OV and elucidated the potential mechanism of the mRLs in immune modulation and treatment response, giving new insights into identifying new therapeutic targets.

**Abstract:**

**Background:** OV is the most lethal gynecological malignancy. M6A and lncRNAs have a great impact on OV development and patient immunotherapy response. In this paper, we decided to establish a reliable signature of mRLs. **Method:** The lncRNAs associated with m6A in OV were analyzed and obtained by co-expression analysis of the TCGA-OV database. Univariate, LASSO and multivariate Cox regression analyses were employed to establish the model of mRLs. K-M analysis, PCA, GSEA and nomogram based on the TCGA-OV and GEO database were conducted to prove the predictive value and independence of the model. The underlying relationship between the model and TME and cancer stemness properties were further investigated through immune feature comparison, consensus clustering analysis and pan-cancer analysis. **Results:** A prognostic signature comprising four mRLs, WAC-AS1, LINC00997, DNM3OS and FOXN3-AS1, was constructed and verified for OV according to the TCGA and GEO database. The expressions of the four mRLs were confirmed by qRT-PCR in clinical samples. Applying this signature, one can identify patients more effectively. The samples were divided into two clusters, and the clusters had different overall survival rates, clinical features and tumor microenvironments. Finally, pan-cancer analysis further demonstrated that the four mRLs were significantly related to immune infiltration, TME and cancer stemness properties in various cancer types. **Conclusions:** This study provided an accurate prognostic signature for patients with OV and elucidated the potential mechanism of the mRLs in immune modulation and treatment response, giving new insights into identifying new therapeutic targets.

## 1. Introduction

Ovarian cancer is a gynecological malignancy which is prevalent in women aged 40 to 79 [1]. Additionally, OV is a common type of this cancer, and the occurrence of OV is ninety percent in all patients with ovarian cancer [2]. Due to the lack of specific initial symptoms and sensitive biomarkers for early diagnosis, most OV patients cannot be diagnosed at an early stage and the progression is rapid [3,4]. Consequently, developing novel and reliable signatures to diagnose and prognose OV at an early stage is an urgent need.

The results of randomized trials show that transvaginal ultrasound and CA-125 testing are of little help to diagnose ovarian cancer and change the mortality rate [5].

Therefore, more attention has been given to discovering key regulatory genes involved in cancer pathogenesis and progression and constructing prognostic signatures based on them. Recently, as a reversible epigenetic modification of various RNAs, the vital role and molecular mechanisms of m6A RNA modification in cancer pathogenesis, drug response and developing new targets for therapy have been intensely studied and confirmed [6]. Furthermore, it has also been found that m6A RNA modification functions in immunity, which provides insights into cancer immunotherapy [7]. M6A is dynamically regulated by three types of proteases with distinct functions, namely “writers, “erasers” or “readers” [8,9]. Changing the structure of RNA can influence a variety of cell processes. As a result, the effect of mRLs may play an important role in the transfer of cancer cells.

LncRNA is a group of RNA molecules. Although lncRNA cannot be translated into protein, it has a vital effect on many biological processes such as tumorigenesis and development [10,11,12]. The relationship between m6A and lncRNAs is under in-depth study. Extensive studies have revealed that the biogenesis and functions of lncRNAs depend on m6A modification [13,14,15]. YTHDF1 and YTHDF2, m6A readers, modify the stability of oncogenic lncRNA THOR, thus promoting the biological behavior of cancer cells [16]. LncRNA RP11 dependent on m6A induction can trigger colorectal cancer metastasis by preventing the degradation of Zeb1 [17]. In addition, m6A modification is also adversely affected by lncRNAs [18], and they can function together to regulate protein translation [18,19]. For instance, LINRIS stabilize IGF2BP2 (an m6A reader) by preventing its K139 ubiquitination, thus promoting colorectal cancer cell line growth [20]. In OV, the stability of RHPN1-AS1 is improved by m6A modification, promoting the proliferation and metastasis of OV cell [21]. Hence, linking m6A and lncRNA, and further studying the function and molecular mechanism of mRLs in tumorigenesis and progression can affect the overall survival rate of OV. However, such research on OV is still lacking.

This study established and verified a mRL prognostic signature for OV using the TCGA and GEO database. The underlying correlation between the signature and TME, as well as therapeutic response, was also explored. Moreover, clustering subgroups and pan-cancer analysis were also used to verify the application value of the model in distinguishing patients in terms of prognosis and therapeutic response.

## 2. Materials and Methods

### 2.1. Collecting and Disposing Data

The flow chart of our study is performed in Figure 1a. We collected gene expression profiling and clinicopathological data of OV samples from the TCGA dataset (https://cancergenome.nih.gov/, (accessed on 5 February 2022)) and the GSE9891 database (http://www.ncbi.nlm.nih.gov/geo/, (accessed on 5 February 2022)). The single gene expression in OV samples with different stages was obtained from GEPIA (http://gepia.cancer-pku.cn/index.html, (accessed on 5 February 2022)) [22]. To identify the role of the four mRLs in more cancers, we obtained the pan-cancer data of 33 cancers from the UCSC Xena database (https://xenabrowser.net/datapages/, (accessed on 5 February 2022)) [23], including RNA-seq, immune subtypes, prognosis profiles and stemness scores.

### 2.2. Identification of mRLs Associated with OV Prognosis

Firstly, the expression matrix of 23 m6A regulators (METTL3, METTL14, METTL16, WTAP, VIRMA, ZC3H13, RBM15, RBM15B, YTHDC1, YTHDC2, YTHDF1, YTHDF2, YTHDF3, HNRNPC, FMR1, LRPPRC, HNRNPA2B1, IGFBP1, IGFBP2, IGFBP3, RBMX, FTO and ALKBH5) was achieved in previous studies [24]. Secondly, Pearson correlation analysis was implemented to select 419 mRLs according to the standard of *p* < 0.05 and absolute correlation coefficient >0.3, then 61 mRLs shared in both TCGA and GEO were obtained after the removal of batch effects through combat via the “sva” R package. Finally, we collected 374 samples with sufficient prognostic clinical data and lncRNAs expression information from TCGA. The 374 samples from the TCGA-OV dataset were assigned into TCGA-train (*n* = 187) and TCGA-test (*n* = 187) according to 1:1 randomly, and we employed the train set to build the signature. The test set (187 samples), the entire set (374 samples) and the GSE9891 database (278 samples) were applied for validation and further research. Univariate Cox regression analysis was applied to screen the relationships between mRLs related to the patients’ prognoses in the TCGA-train set using the survival package of R (https://www.bioconductor.org/packages/release/bioc/html/rbsurv.html, (accessed on 5 February 2022)).

### 2.3. RNA Isolation and qRT-PCR

The research was approval by the First Affiliated Hospital of Nanjing Medical University Ethics Committee and the participants who took part in the study all provided written informed consent. Total RNA of the OV and normal tissues was obtained through TRIzol reagent (Thermo Fisher Scientific, Waltham, MA, USA), whose integrity was estimated by the Agilent Bioanalyzer 2100 (Agilent Technologies, Santa Clara, CA, USA). cDNA synthesis was conducted using the high-capacity reverse transcription kits (TaKaRa, Shiga, Japan) and then qRT-PCR was based on SYBR Green PCR Kit (Thermo Fisher Scientific, Waltham, MA, USA) and the 2^−ΔΔCt^ method on Light Cycler 480 (Roche, Switzerland). GAPDH was the endogenous control. Primer sequences for GAPDH and four mRLs were presented in Appendix A.

### 2.4. Establish and Prove an mRL Prognostic Signature for OV

The model was established using LASSO in TCGA-train. Then, the multivariate regression analysis was conducted to determine mRLs with independence. The format to assess the risk score was: $\mathrm{risk}\;\mathrm{score}\;=\sum_{i=1}^{n}\mathrm{coef}\left(i\right)*\mathrm{lncRNA}\left(i\right)\mathrm{expression}$. The patients were divided into two groups in terms of median risk score of train set. Kaplan–Meier analysis of the OS of the groups was conducted through the “survival” R package. Through the “ROC” R package, the ROC was depicted and the AUC was calculated to determine its specificity and sensitivity. PCA was used to prove the grouping ability of the model by grouping the visualization of high-dimensional data of the risk model [25]. We also conducted subgroup analysis to verify the prognostic ability of the mRL model. Univariate and multivariate Cox analysis were applied to assess whether the signature was an independent factor.

### 2.5. Establishment and Validation of a Nomogram

To improve the predictive ability of the mRL signature, we established a nomogram composed of the signature, patient age, grade and tumor stage. Calibration curves were applied to explore the accuracy and reliability of the nomogram.

### 2.6. Gene Set Enrichment Analysis

As carried out in our previous study [26], we performed function annotation through GSEA to reveal potential mechanisms in different risk groups. KEGG in GSEA was applied to select predefined gene sets; 5000 permutations were conducted for the gene set to calculate *p*-values. GSEA was analyzed using “clusterProfiler” package [27].

### 2.7. The Correlation between the Signature and Tumor Infiltration Immune Cells (TIICs)

CIBERSORT was employed to assess the abundance of TIIC profiles in all tumor tissues. We assessed the proportion of 22 types of TIICs in every patient according to the CIBERSORT score [28]. Furthermore, the correlation between the lncRNAs and TIICs score was evaluated to identify which immune cells had distinguished between cancers.

### 2.8. The Correlation to Other Immune Features

ESTIMATE was used to compute the proportion of immune and stromal components in TME for each sample [29]. The relationships between lncRNA expression and the risk score were calculated by Spearman correlation. Furthermore, we identified six immune types and used the analysis of variance (ANOVA) to evaluate the relationships between these subtypes and lncRNA expression. The data of cancer stem cell-like properties were employed to assess the stemness characteristics of the cancer cells. The association between stemness features and lncRNA was also evaluated by Spearman analysis.

### 2.9. Immunotherapy Response Prediction

Immunophenoscores (IPSs) are calculated based on four major categories of gene expression z-scores to evaluate and compare the potential reaction to ICI between the two groups, and the high scores represent high immunogenicity [30]. IPS data of each OV patient in TCGA-entire were achieved from The Cancer Immunome Atlas (TCIA) (https://tcia.at/home, (accessed on 5 February 2022)).

### 2.10. Assess Drug Sensitivity

The newest CellMiner version (https://discover.nci.nih.gov/cellminer/, (accessed on 5 February 2022)) could help researchers achieve the NCI-60 data flexibly for correlations between genomic, molecular and pharmacologic parameters [31]. We collected the expression of 4 m6A-related genes and z-score for cell sensitivity data (GI50) from the webpage and used Pearson’s correlation coefficient to assess the influence of the m6A-related gene expression on drug sensitivity.

### 2.11. Consensus Clustering

The expression of the lncRNAs was utilized to recognize the subtypes in OV patients via “ConsensusClusterPlus” R package (http://www.bioconductor.org/, (accessed on 5 February 2022)). We used the log-rank test and K–M curve to calculate the OS difference between groups.

### 2.12. Statistical Analysis

R 4.0.2 was used to carry out statistical analyses. The relationships between the risk score or the mRL expression level and stemness score, estimate score and drug sensitivity were calculated using Spearman or Pearson correlation analysis. Univariate or multivariate regression analyses were applied to calculate the relationships between the four mRLs with patient OS. Kaplan-Meier analysis could compare the survival state between the groups. Time-dependent ROC curve analysis could test the predict ability of the model. Subgroup analysis could evaluate the stability of the prognostic signature in subgroups stratified by clinical characteristics. Student’s *t*-test and ANOVA were applied to find the differences between the different groups. Linear mixed-effect models were used for pan-cancer analysis. The hazard ratio and 95% confidence interval were assessed to screen prognosis-related genes. *p* < 0.05 was regarded as statistically significant.

## 3. Results

### 3.1. Identification of mRLs in OV

Firstly, we abstracted the expression of lncRNAs from the TCGA-OV dataset and identified 419 mRLs (*p* < 0.05, absolute correlation coefficient > 0.3) through Pearson’s correlation analysis with m6A regulators. The relational network between the m6A regulators and mRLs was shown in Figure 1b. After collecting the expression of lncRNAs in the GSE9891 dataset, we found 61 mRLs shared in both datasets. When combined with clinical information, we screened five mRLs related to prognosis through the univariate regression analysis of the TCGA-train (Appendix A).

### 3.2. Establish and Verify a Prognostic Signature Based on mRLs in OV Patients

Firstly, 374 OV patients were randomly classified into train and test sets at a 1:1 ratio, including 187 OV patients, respectively. Secondly, we performed the LASSO (Appendix A) and multivariate regression analysis (Appendix A) in the train set to build a reliable prognostic signature. The formula was as follows:

Risk core = (−0.068088177 * expr (WAC-AS1)) + (−0.276777737 * expr (LINC00997)) + (0.09155959 * expr (DNM3OS)) + (−0.137781856 * (FOXN3-AS1), expr means the expression value. WAC-AS1, LINC00997, DNM3OS and FOXN3-AS1 were finally identified as key mRLs. Both the outcomes of the expression analyses in the GEPIA dataset and the qRT-PCR results indicated that the expression of DNM3OS and FOXN3-AS1 was higher in normal tissue compared to the tumor tissue (*p* < 0.05) (Appendix A). Afterward, we assessed the risk score of OV patients in the TCGA-train and separated them into two groups. The distributions of risk score, survival status of each sample (Figure 2a) and the heatmap of key gene expression patterns (Figure 2e) in the TCGA-train are shown in Figure 2. People in the high-risk group had poor survival prognosis compared to the low-risk groups in the TCGA-train (Figure 2i). Time-dependent receiver-operating characteristic curve (ROC) analysis based on the signature revealed that the AUC of 1-, 2- and 3-year survival was 0.624, 0.694 and 0.630, respectively (Figure 2m). To verify the mRL prognostic signature, we calculated the risk score of the patients in the TCGA-test (N = 187), TCGA-entire (N = 374) and GEO datasets (N = 278) and divided the patients into either group according to their median risk score of the TCGA-train (Figure 2b–d). The heatmap showed that, with the exception of DNM3OS, the expression of WAC-AS1, LINC0099 and FOXN3-AS1 was higher in the low-risk groups (Figure 2e–h). The Kaplan–Meier analyses of the TCGA-test, TCGA-entire and GEO datasets revealed the same trend, that the OV patients with a low risk had higher OS than the high-risk group (*p* < 0.01) (Figure 2j–l). Furthermore, the time-dependent ROC curve highlighted the capability of the prognostic signature based on mRLs. The AUC of 1-, 2- and 3-year survival in the three sets is shown in Figure 2n–p. In addition, the distributions of the groups are different in the PCA plots (Appendix A). These results all demonstrate the accuracy and reliability of the mRL prognostic signature in OV prediction.

### 3.3. The Relationships between the Model and Clinical Factors

The correlations between the risk core and clinical characteristics are performed in Appendix A. The risk score in the age ≤ 60 group was lower than the age > 60 group (*p* < 0.05, Appendix A), and the proportion of older patients with a high-risk score was 52%, while the people with a low-risk score was 40% (Appendix A). However, the difference in the risk score in the groups assigned by tumor grade and tumor stage was not significant (Appendix A). Furthermore, the risk score of the high immune score group was markedly higher than the lower score patients (*p* < 0.05, Appendix A), and the significant difference could be found in the risk core of Cluster 1 and Cluster 2 subgroups (*p* < 0.05, Appendix A). Stratification analysis grouped by patient age, tumor grade and tumor stage highlighted that the OS of the patients in the high-risk group was worse than the low-risk group (Appendix A). When comparing the patients’ states between the groups, we also found a higher mortality of patients in the high-risk group (Appendix A), and the risk scores of dead patients were statistically higher than the alive patients (*p* = 0.002, Appendix A). K–M curves of OS for patients subjected to chemotherapy demonstrated that patients at a high risk had poorer prognoses than those at a lower risk (*p* < 0.001, Appendix A). In patients with BRCA1, the low-risk group also had a favorable survival outcome (Appendix A).

### 3.4. Clinical Application of the Signature

Univariate and multivariate regression analyses in the TCGA-train, TCGA-test, TCGA-entire and GEO datasets all indicated the independence of the mRL model (Appendix A). Moreover, patient age and tumor stage were also crucial poor prognostic factors for OV. To predict performance of the signature, we built a nomogram comprising the risk score, age, grade and stage in both the TCGA (Figure 3a) and GEO datasets (Appendix A). Figure 3b–d revealed that the predicted rates of OS were highly consistent with the observed rates. In addition, the prognostic signature showed superior predictive ability compared to other clinical characteristics, and when combined with clinical characteristics, the model showed better predictive power than the signature when used alone (Figure 3e).

### 3.5. The Correlations between the mRL Prognostic Signature and TME

GSEA indicated that the pathway related to immunity was enriched in people with a high-risk score (Figure 4a,b). Firstly, we implemented the ESTIMATE algorithm which indicated that the stromal score, immune score and ESTIMATE score were higher in the high-risk group (*p* < 0.05), and the risk score had a positive association with them (Figure 4c–h). The same trends were found in the GEO dataset (Figure 4i–n). Then, by comparing the 22 TIICs between the groups, we identified that the resting dendritic cells and M2 macrophages were positively related to the risk score and (*p* < 0.05) and the infiltration level of the T follicular helper cells and regulatory T cells (Tregs) had a negative correlation with the risk score (*p* < 0.05) in the TCGA dataset (Figure 5a–i). In the GEO dataset, the risk score had a positive correlation with gamma delta T cells and resting memory CD4 T cells, while the risk score has a negative relationship with Tregs, resting NK cells and activated dendritic cells (Appendix A). The differences in immune score and abundance of TIICs between the groups showed that patients with a high risk had a repressive immune phenotype, which could partly explain the poorer OS of people with a high risk score.

To identify the immunological role of the four mRLs, we assessed their association with the TIICs using the CIBERSORT algorithm. We found that DNM3OS had a significant association with T follicular helper cells, M2 macrophages and memory B cells (*p* < 0.001); WAC-AS1 was correlated with T follicular helper cells and M2 macrophages (*p* < 0.001) (Appendix A). Moreover, WAC-AS1 (R = −0.15, *p* = 0.0041) and LINC00997 (R = −0.23, *p* < 0.001) had negative relationships with the immune score, whereas DNM3OS (R = 0.13, *p* = 0.008) and FOXNS-AS1 (R = 0.19, *p* < 0.001) were positively related to the immune score (Appendix A). Next, to clarify the potential mechanisms of the lncRNAs in tumor progression, we used GSEA analysis to establish the enriched pathways of the mRLs in OV (Appendix A). DNM3OS was highly enriched in the calcium signaling pathway, the focal adhesion pathway, the hematopoietic cell lineage pathway, the neuroactive ligand–receptor interaction pathway and cancer pathways.

The expression level of PD-L2 was higher in tumor tissues than in normal tissues (*p* < 0.05, Figure 6a), and it was markedly higher in people with a high risk score in both the TCGA and GEO dataset (*p* < 0.05, Figure 6c–f), suggesting that people in the high-risk group will benefit less from immune checkpoint inhibitors (ICIs). In addition, LINC00997 showed a high correlation with PD-L2 (Figure 6b); thus, the function of LINC00997 in the tumor immune microenvironment deserves further research. Comparing the IPSs between the groups, low-risk groups had significantly higher IPSs, suggesting a more immunogenic phenotype (Figure 6g–j). These results all demonstrate that the signature can predict the efficacy of immunotherapy for OV patients.

### 3.6. Consensus Clustering for mRLs Related to OV Prognosis and TME

After the consensus clustering, OV patients in the TCGA dataset were classified into Cluster 1 and Cluster 2 subgroups following the expression of the significant prognostic mRLs (Figure 7a–d). The heatmap of the expression pattern between Cluster 1 and Cluster 2 subgroups was performed in Figure 7e. In addition, the stroma, immune and ESTIMATE scores were markedly higher and the tumor purity was substantially lower (*p* < 0.05) in Cluster 2 (Figure 7g–j). Moreover, the infiltration of naïve B cells (*p* < 0.05), T cell activated memory CD4 cells (*p* < 0.05), T follicular helper cells (*p* < 0.05), M1 macrophages (*p* < 0.05) and resting mast cells (*p* < 0.05) was higher in the Cluster 1 subgroup, whereas the abundance of M0 macrophages (*p* < 0.05), M2 macrophages (*p* < 0.05), activated mast cells (*p* < 0.05) and neutrophils (*p* < 0.05) was higher in the Cluster 2 subgroup (Figure 7f); thus, the Cluster 2 subgroup tended to behave more as immunosuppressive phenotype. Moreover, the results of Kaplan–Meier analyses highlighted that the OS of Cluster 2 was dramatically worse than that of Cluster 1 (Figure 7k), and the expression of PD-L2 was higher in Cluster 2 (Figure 7l); this indicates that patients in the Cluster 2 may respond more sensitively to immunotherapy. The above results all demonstrate that the expression of the mRLs influences the TME, leading to different prognoses.

### 3.7. The Expression and Immune Status of lncRNAs in Pan-Cancer

We used the pan-cancer data of 33 cancers from TCGA to further study the relationships between the four mRLs and immune features, stem-like properties and patient prognoses in pan-cancer. The expression pattern of lncRNAs in 33 cancers (Figure 8a–d) and the heatmap (Appendix A) showed inter-tumor heterogeneity, but in most cancers, the expression level of WAC-AS1 was higher in tumor tissues, whereas the expression level of DNM3OS was the opposite, which was consistent with the expression trend in OV (Appendix A). The expression distribution of the four mRLs across all 33 cancer types showed the same trend as the above results (Figure 8e). Moreover, there is no significant correlation between the key mRLs (Appendix A). The Kaplan–Meier analyses (Appendix A) and univariate regression (Appendix A) analyses showed that the role of the mRLs in the survival of patients with different cancers varied.

To identify the relationships between the key mRLs and immune features in pan-cancer, we investigated the immune subtype and the landscape of correlation with stromal score, immune score, stemness scores based on DNA-methylation (DNAss) and stemness scores based on mRNA (RNAss). Through an extensive immunogenomic analysis of pan-cancer, six immune subtypes, including wound healing (C1), IFN-γ dominant (C2), inflammatory (C3), lymphocyte depleted (C4), immunologically quiet (C5) and TGF-β dominant (C6), were identified according to differences in immune, genetic and clinical features, in which C3 had great OS, C2 and C1 had poor prognosis, while C4 and C6 had the least favorable outcomes [32]. We evaluated the correlations between the mRL expression level and the six immune subtypes and found that all of them were related to pan-cancer immune subtypes and elevated C1, C2 and C6 subtypes with upregulated expression of DNM3OS (Figure 9a). Additionally, DNM3OS had a positive correlation with stromal score and immune score, whereas WAC-AS1 and LINC00997 were negatively related to stromal and immune score (Figure 9b–d), which showed similar correlation in OV. All four mRLs were negatively related to RNAss (Figure 9e). Therefore, the four mRLs took part in the formation of specific TIME. Finally, we applied the CellMiner database to evaluate the correlation between the mRLs and drug sensitivity. A higher z-score means higher sensitivity to one specific drug. Significantly, the expression of FOXNS-AS1 influenced drug sensitivity in various cell lines (Appendix A).

## 4. Discussion

Due to the rapid progression and challenge of early diagnosis, people with OV face a poor OS, with five-year survival rates below 45% [33]. More and more efforts have been carried out to discover robust and sensitive predictive models for early diagnosis and prognosis prediction, including models based on immune genes, autophagy-related genes, m6A regulators and so on [34,35,36,37,38]. We found emerging evidence proving that m6A modification had a crucial effect in the formation of TME landscape heterogeneity and complexity [39,40,41]. Zhang et al. constructed an m6A score and demonstrated that a low m6A score was correlated with a high mutation burden, an inflamed TME phenotype and improved survival [42]. As the main modification for RNAs, m6A modifications almost regulate all biological pathways and their changes in tumors reprogram the tumor immune microenvironment, helping tumor cells escape and metastasize [15,43]. This study focused on mRLs, established a signature for OV patients and validated its predictive capability using TCGA and GEO datasets.

The model was established based on four mRLs associated with patient prognoses, including DNM3OS, WAC-AS1, FOXNS-AS1 and LINC00997, among which DNM3OS was correlated with a high-risk score, while the others were correlated with a low-risk score. It was proved that the overexpression of DNM3OS played a role in the development of ovarian cancer [44]. WAC-AS1, a competing endogenous RNA, has a great influence on the regulation of tumor glycolysis [45]. Additionally, the prognoses of glioma patients who had low WAC-AS1 expression were better than the high-expression group [46]. In contrast to our results, it was reported that LINC00997 had a positive relationship with the metastasis and development of colorectal cancer [47]. Subsequently, OV patients were separated into two groups in terms of the median risk score, and the patients in higher risk group had a worse prognosis. Multivariate Cox regression analysis highlighted that the model had a great effect on OS. ROC analysis proved that the model was more accurate than single clinical characteristics to predict the outcomes of OV. We constructed a nomogram for clinical application. GSEA revealed that people with a high-risk score were significantly enriched in immune response and immune system process pathways. We believe that the mRLs can take part in the mechanisms of tumorigenesis and progression of OV, and the signature might provide an accurate prediction and a theoretical foundation for the treatment of OV.

Due to the resistance to chemotherapy, patients with OV ultimately die of recurring lesions after surgical removal. To improve the prognoses of OV patients, new treatment modalities have emerged, especially immunotherapy, which could boost the patient’s immune system to find and kill tumor cells. The immune landscape of an OV patient determines their specific response to various treatments and subsequent prognosis. The proportion of intratumoral CD8+TILs, CD4+ Treg, macrophages and MDSC was significantly correlated with patients’ outcomes [48,49,50]. This finding has led to several stratification strategies, such as based on the number of CD8+TILs [50], six immune subtypes [32] or cytotoxic immunophenotypes [51] to identify those who benefit from immunotherapy. After validating the mRL prognostic signature in TCGA and GEO datasets, we performed immune subtype, consensus clustering and pan-cancer analyses to further research their function when regulating TME and the underlying molecular mechanism of interaction between lncRNA and m6A regulators in OV.

TIME is associated with cancer occurrence and progression [48,52]. This study calculated the correlation between the signature and TME in OV. According to the above results, high immune scores means higher risk scores. Some published articles showed that OV higher immune score patients had a poorer prognosis [53,54], consistent with our findings. After that, positive associations were identified between the risk score and resting dendritic cells, M2 macrophages, gamma delta T cells as well as resting memory CD4 T cells, while negative associations were identified between the risk score and T follicular helper cell infiltration, Tregs and resting NK cells. TIICs in the TME have been found to be related to the prognoses of OV patients [55]. For example, tumor-infiltrating macrophages with an M2 phenotype showed immunosuppressive activity, negatively related to OS [56]. It was also reported that the function of resting dendritic cells could be inhibited through GDF-15 interacting with CD44, thus facilitating ovarian cancer immune escape [57]. Therefore, the high level of M2 macrophages is closely correlated with a higher risk score and poorer prognosis, while resting NK cell infiltration improved the survival of patients with ovarian cancer [58,59]. Therefore, bad clinical prognosis of high-risk patients may be associated with TIICs, and mRLs may act as significant regulators of TIIC infiltration in OV.

Cancer cells can create an immunosuppressive microenvironment by activating immune checkpoint pathways, thus achieving immune escape [60]. Therefore, blocking the role of immune checkpoints in cancer treatment has attracted attention. Some ICIs have been approved for clinical application in several human malignancies [61]. Although no study has found an improvement in the survival of patients in clinical trials, including using ICIs targeting PD1, PD-L1 and CTLA4 in OV [62], identifying treatment targets, predicting treatment response, screening potential drugs and providing new immunotherapy will provide new breakthroughs for combined and personal therapy of OV [63]. The relationships between mRLs and immune checkpoint expression indicated that the risk score had a positive relationship with the PD-L2 expression level. The high expression of an immune checkpoint ligand PD-L2 represents a suboptimal response to ICIs [64]. The low-risk groups exhibited higher IPSs, indicating they may benefit more from immunotherapy, which is consistent with previous research that found that the anti-tumor responses mediated by ICIs depend on basic immunophenotypes [65,66,67,68]. Based on the above results, we propose that our risk signature may help predict immunotherapy efficacy for OV. Additionally, our study highlights the biological mechanism of mRLs in the occurrence and progression of OV.

Furthermore, we conducted consensus clustering analysis to screen m6A-related genes of OV and achieved two clusters according to the expression of four mRLs. We found that the expression of WAC-AS1, LINC00997 and FOXN3-AS1 was higher, and the expression of DNM3OS was lower in Cluster 1; thus, patients in Cluster 1 had lower risk scores. We compared the biological characteristics between the two clusters to further clarify the relationships between immune features and the mRLs. Cluster 1 showed higher infiltration of niave B cells, memory-activated CD4 T cells, T follicular helper cells and M1 macrophages than Cluster 2, while Cluster 2 had a higher infiltration of M2 macrophages than Cluster 1. The estimate score was higher in Cluster 2, and the tumor purity was higher in Cluster 1, which was consistent with the above results. The Kaplan–Meier analysis suggested that Cluster 2 had a poor prognosis. Furthermore, the expression of PD-L2 was higher in Cluster 2, which suggested that Cluster 1 respond more sensitively to ICIs. How m6A modification influences the tumor microenvironment, especially the TIIC types, has been researched [42,69,70]; our study might provide more evidence for further research.

Finally, we used pan-cancer analysis to further study the role of the mRLs in different cancers. The expression pattern of the four mRLs in pan-cancer showed intertumor heterogeneity, but in most cancer types, WAC-AS1 had a higher expression in tumor tissues, whereas the expression level of DNM3OS was the opposite, which was consistent with the expression trend in OV. By calculating the correlation between the lncRNAs and immune features, we found that DNM3OS was closely related to M2 macrophages (*p* < 0.001) in OV, elevated C1, C2 and C6 subtypes and an immune score with upregulated expression of DNM3OS in pan-cancer. Das et al. proved that the overexpression of DNM3OS promoted macrophage inflammatory phenotype, immune response genes and phagocytosis [71]. It was also reported that the function of DNM3OS was cooperated by miR-214 through the proinflammatory TLR4/IFN-γ/STAT1 pathways regulated pericystic macrophage accumulation [72]. These findings all demonstrate that mRLs play a role in TME and significantly influence OS.

Our study still has several limitations. Firstly, the results were entirely based on open access databases, limiting the validation of our cohort and external multicenter cohorts. Secondly, the interactions between mRLs and m6A regulators and the mechanism of the function of lncRNAs in TME were not demonstrated by the experiments. Thirdly, the prognostic signature needs to be applied in a real-world setting in order to explore its sensitivity and efficiency in classifying patients for target therapies.

## 5. Conclusions

In conclusion, our study established an independent and reliable prediction signature of mRLs, and systematically evaluated its predictive accuracy and role in TIME. Consensus clustering analysis and pan-cancer analysis demonstrated the vital role of the specific mRLs in different cancer types, which may help the discovery of novel therapeutic targets for OV.

## Figures and Tables

**Figure 1 cancers-14-04056-f001:**
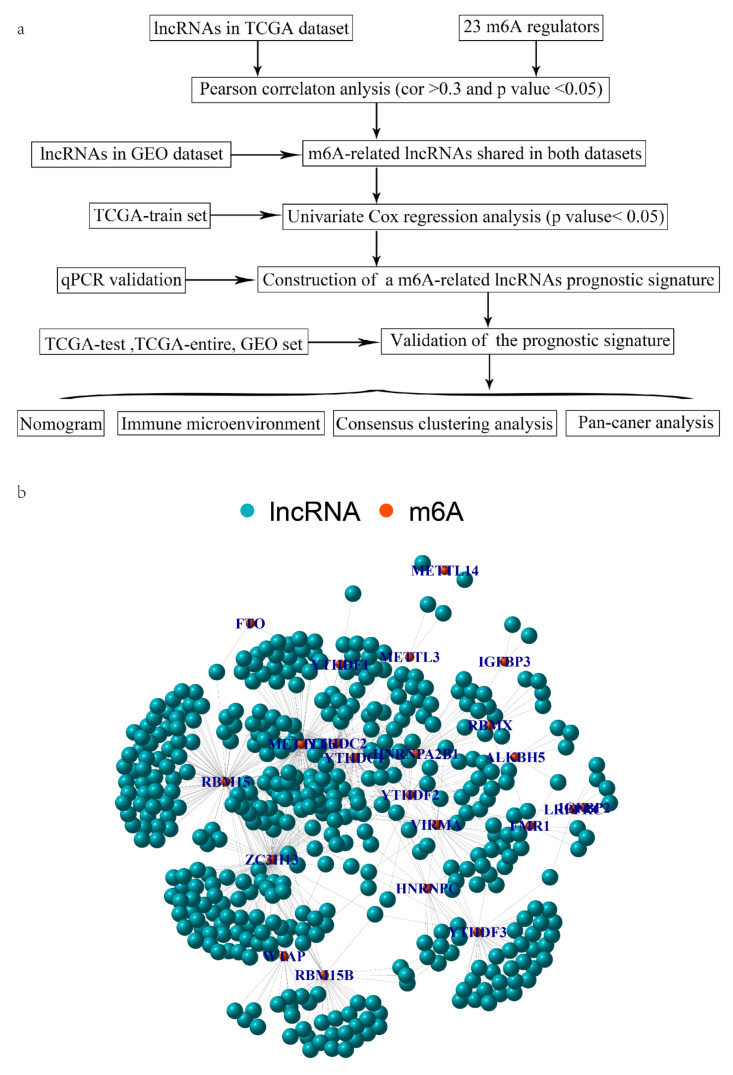
(**a**) Flow chart of the study and (**b**) co-expression network diagram of m6A regulators and lncRNAs in the TCGA dataset.

**Figure 2 cancers-14-04056-f002:**
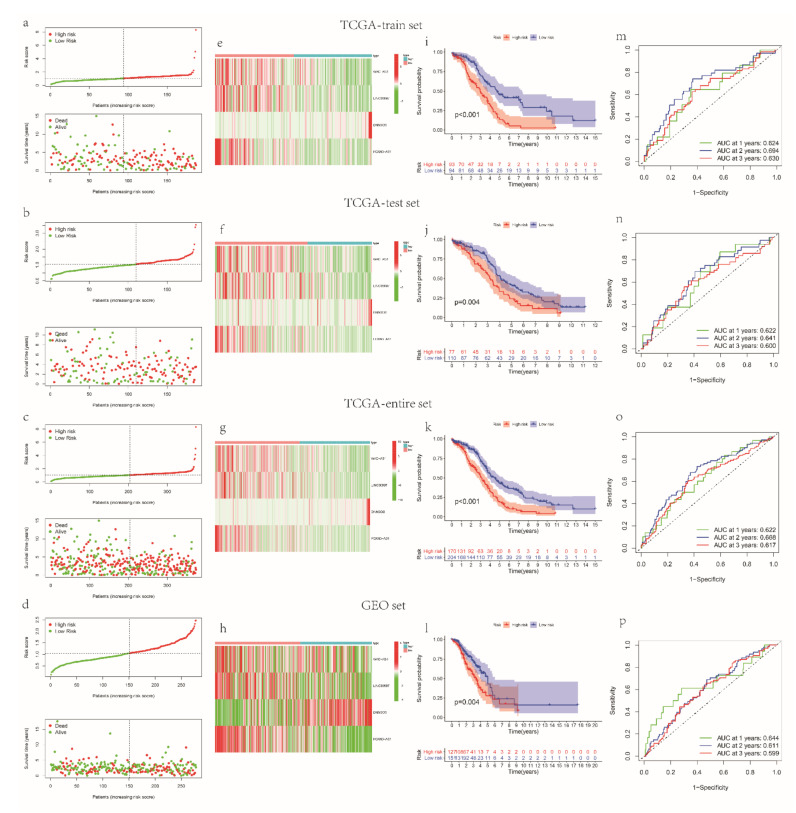
Construction and validation of the mRL prognostic signature. Distribution of risk score, OS status (**a**–**d**) and heatmap of the expression of the four prognostic mRLs (**e**–**h**) in the TCGA-train, test, entire and GEO dataset. (**i**–**l**) Kaplan–Meier curves of OS for OV patients with a high- or low-risk score in the TCGA-train, test, entire and GEO dataset. (**m**–**p**) Time-dependent ROC analysis of risk score in predicting the prognosis in the TCGA-train, test, entire and GEO dataset. ROC, receiver-operating characteristic. In the TCGA-train set, N = 187; in the TCGA-test set, N = 187; in the TCGA-entire set, N = 374; in the GEO set, N = 278.

**Figure 3 cancers-14-04056-f003:**
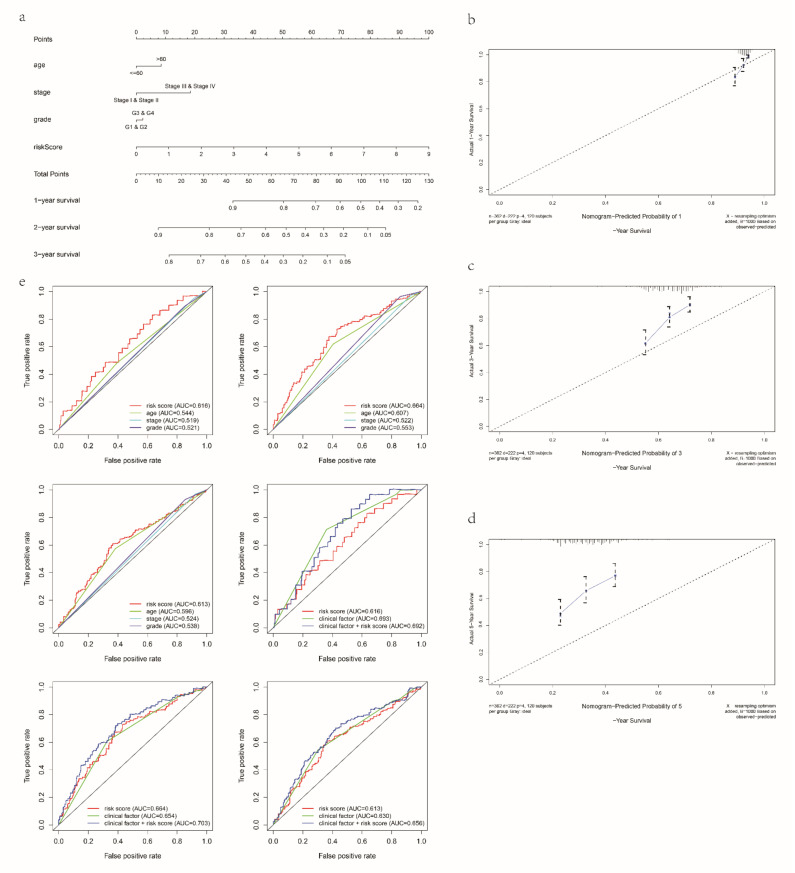
A nomogram for OV patients in TCGA dataset. (**a**) A nomogram for predicting the 1-, 2- and 3-year OS of OV patients. (**b**–**d**). Calibration curves for the prediction of 1-, 2- or 3-year overall survival of OV patients. (**e**) Time-dependent ROC curves for each parameter and the combination of independent prognostic risk factors in the TCGA dataset (for predicting 1, 2 and 3-year OS). In the TCGA-entire set, N = 374.

**Figure 4 cancers-14-04056-f004:**
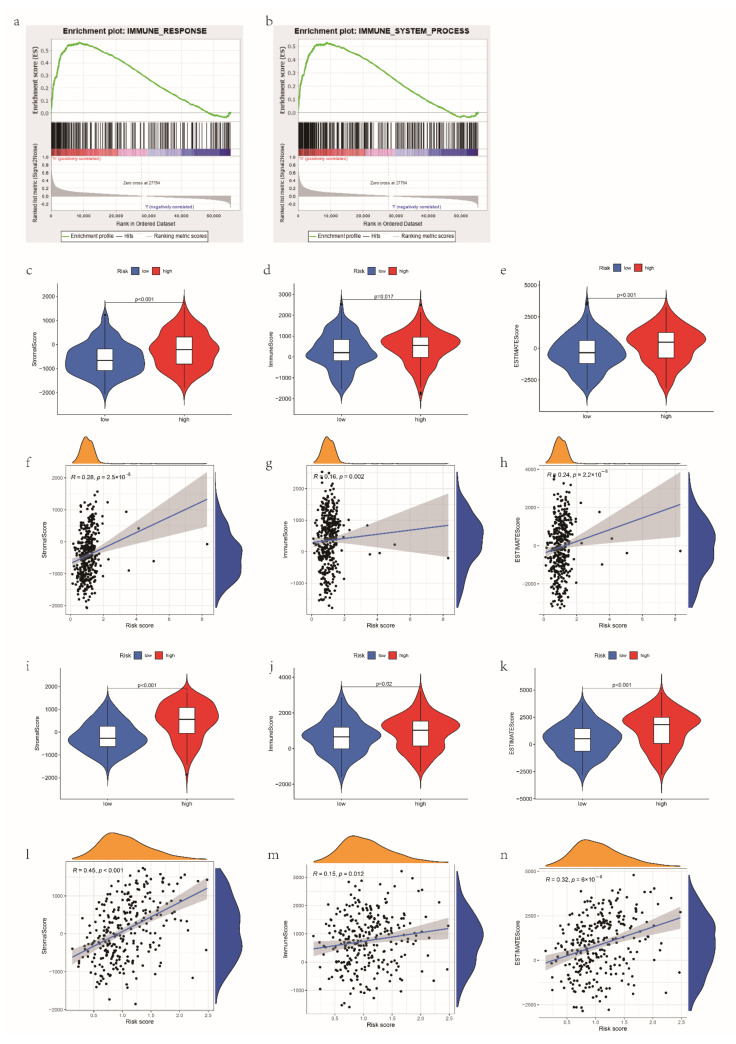
The correlation between risk score and immune features in the TCGA dataset. (**a**,**b**) GSEA showed that immune response and immune system process were enriched in the high-risk group. (**c**–**e**) Stroma, immune and ESTIMATE scores in the high- and low-risk groups in TCGA dataset. (**f**–**h**) The correlation between risk score and stroma, immune and ESTIMATE scores in TCGA dataset. (**i**–**k**) Stroma, immune and ESTIMATE scores in the high- and low-risk groups in GEO dataset. (**l**–**n**) The correlation between risk score and stroma, immune and ESTIMATE scores in GEO dataset. In TCGA-entire set, N = 374; in GEO set, N = 278.

**Figure 5 cancers-14-04056-f005:**
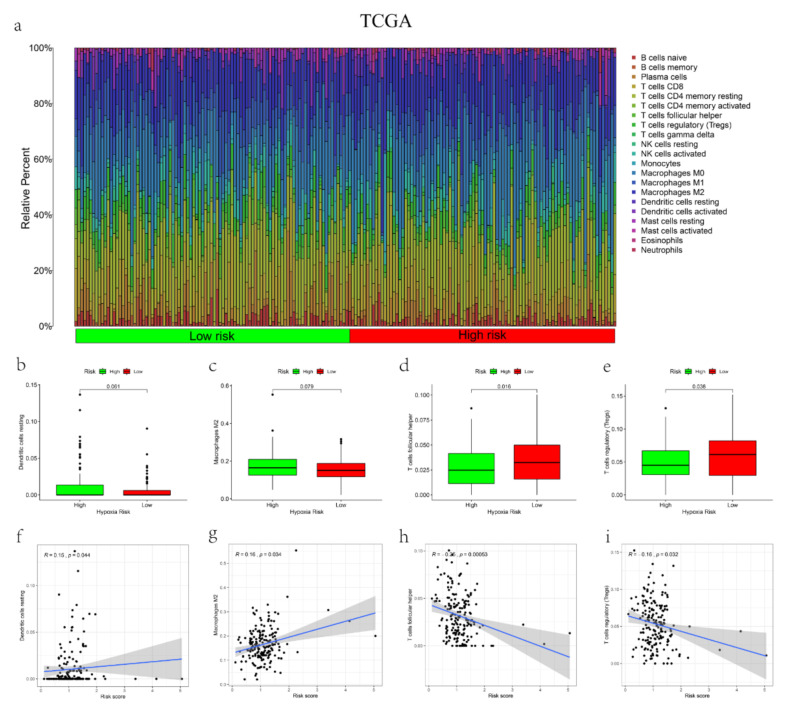
Relationships between the risk score and immune cell infiltration and the role of the mRLs in tumor microenvironment in TCGA dataset. (**a**) The proportion of 22 immune cells infiltration in high- and low-risk groups. (**b**–**i**) Correlation of expression of the mRLs and infiltration of specific immune cell type. In TCGA-entire set, N = 374.

**Figure 6 cancers-14-04056-f006:**
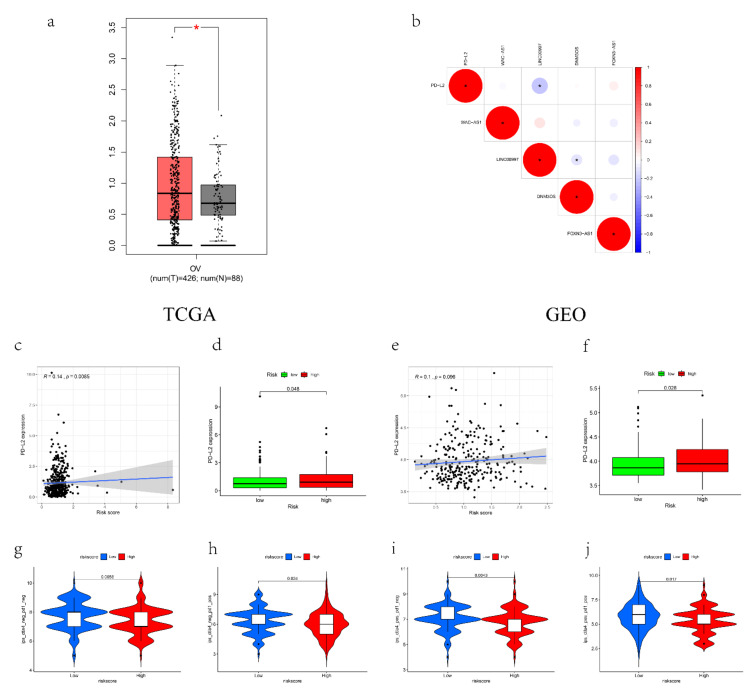
IPSs and immunotherapy gene expression analysis. (**a**) The expression of PD-L2 in tumor and normal tissues. (**b**) The correlation between the mRLs and PD-L2. (**c**–**f**) The expression of PD-L2 in low- and high-risk groups in TCGA and GEO datasets. (**g**–**j**) The IPSs of low- and high-risk groups. * *p* < 0.05, ** *p* < 0.01 and *** *p* < 0.001. In TCGA-entire set, N = 374; In GEO set, N = 278.

**Figure 7 cancers-14-04056-f007:**
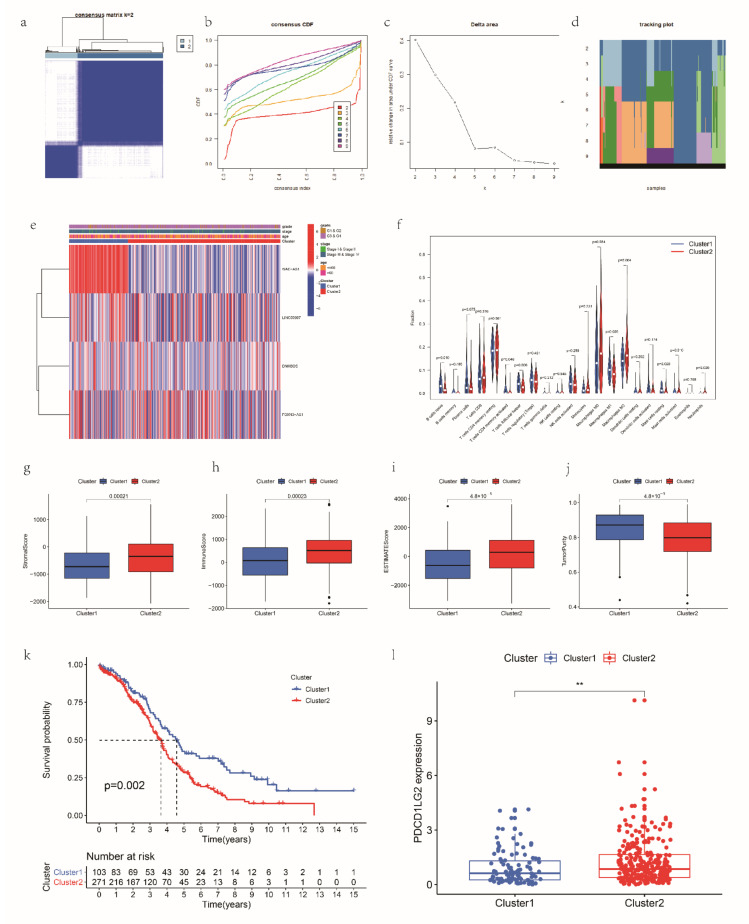
Prognosis and immune infiltrations in consensus clustering subgroups of OV. (**a**) Consensus clustering matrix for k = 2. (**b**) Consensus clustering cumulative distribution function (CDF) for k = 2 to 9. (**c**) Relative change in area under the CDF curve for k = 2 to 9. (**d**) Tracking plot for k = 2 to 9. (**e**) Expression pattern of the mRLs in Cluster 1 and Cluster 2 subgroups. (**f**) The abundance of 21 immune cells in Cluster 1 and Cluster 2 subgroups. (**g**–**j**) Stroma, immune and ESTIMATE scores and tumor purity in Cluster 1 and Cluster 2 subgroups. (**k**) Kaplan–Meier analysis of patients in Cluster 1 and Cluster 2 subgroups. (**l**) The expression of PD-L2 in Cluster 1 and Cluster 2 subgroups.

**Figure 8 cancers-14-04056-f008:**
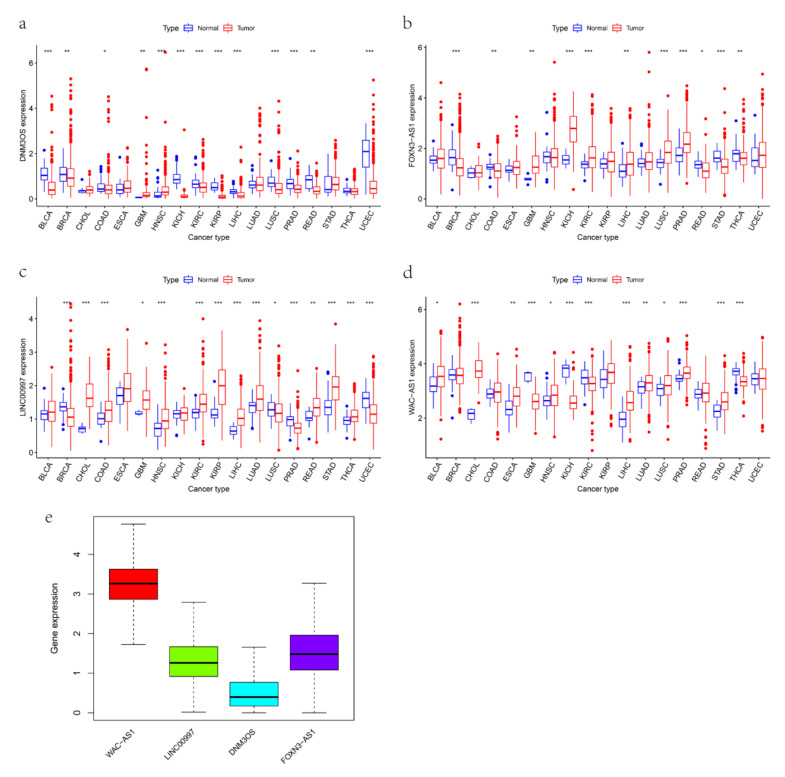
Expression of pan-cancer mRLs. (**a**–**d**) Boxplot showing the expression level of the mRLs in tumor tissue compared with normal tissue in 18 cancer types which were composed of at least 5 normal samples. (**e**) Boxplot showing the expression distribution of the mRLs across pan-cancer. * *p* < 0.05, ** *p* < 0.01 and *** *p* < 0.001.

**Figure 9 cancers-14-04056-f009:**
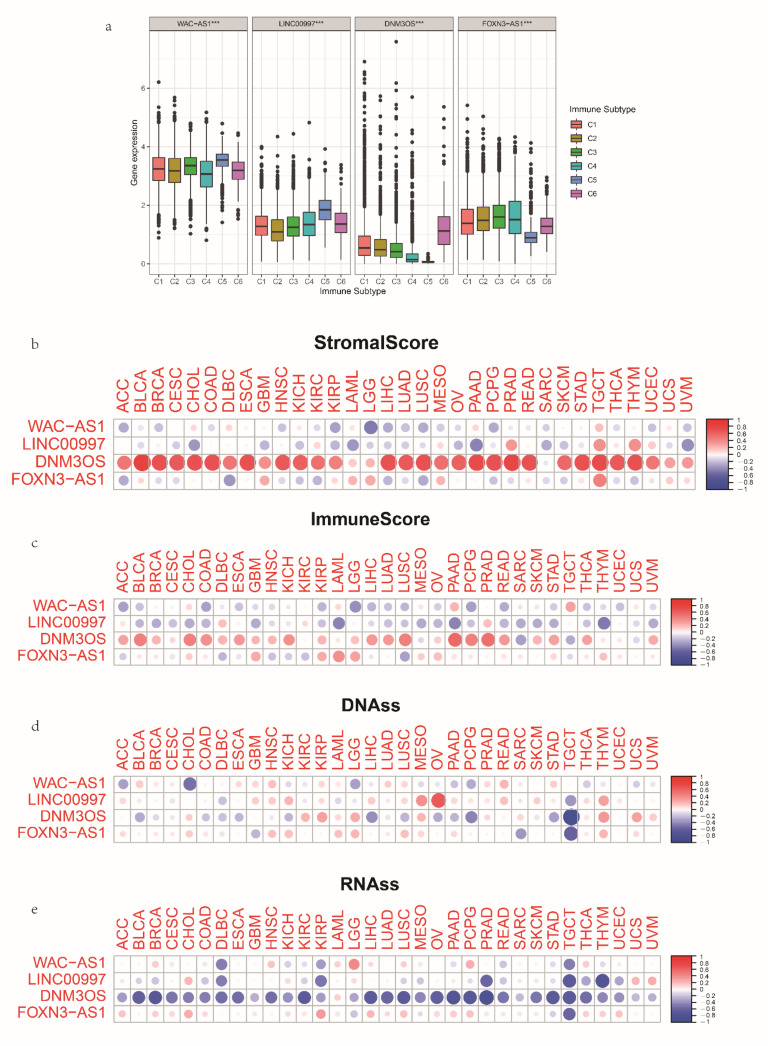
Association of expression of the mRLs with immune subtypes, tumor microenvironment and cancer stemness in pan-cancer. (**a**) Association of expression of the m6A-related lncRNA with immune infiltrate subtypes across all the cancer types tested with ANOVA. (**b**,**c**) Correlation matrix between tumor microenvironment stromal scores (**b**) and immune scores (**c**) and the mRL expression using the ESTIMATE algorithm. (**d**,**e**) Correlation matrix plots showing the association between the m6A-related gene expression and cancer stemness DNA score (**d**) and RNA score (**e**). * *p* < 0.05, ** *p* < 0.01 and *** *p* < 0.001.

## Data Availability

The datasets supporting the conclusions of this article are included within the article.

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
