# Peer review of "The m6A-Related Long Noncoding RNA Signature Predicts Prognosis and Indicates Tumor Immune Infiltration in Ovarian Cancer"

_cancers, 2022, doi:10.3390/cancers14164056_

Round 1

Reviewer 1 Report

The article seems relevant to us in the search for predictive models in patients with ovarian cancer. This line of individualization and personalization of diagnoses, treatment and prognosis is part of personalized medicine.

In light of the results to clinical researchers, we miss the proposal of lines of applicability to daily practice.

Algorithms that order and differentiate therapeutic actions in the different patient profiles.

We believe that the article would complement a proposal for action and integration of the results in the management of patients.

Author Response

Special thanks to you for your good comments. OV is the most lethal gynecological malignancy. M6A and lncRNAs have great influence on OV development and patients' immunotherapy response. Here, we provided an accurate prognostic signature for patients with OV and elucidated the potential mechanism of the N6-methyladenosine related long-noncoding RNAs in immune modulation and treatment response, giving new insights into identifying new therapeutic targets.

Reviewer 2 Report

The authors build a prognostic risk model for ovarian cancer based on
4 long-noncoding RNAs selected among lncRNAs correlated with
expression of m6A regulators. Furthermore, the authors correlate the
risk score with several other samples' features such as gene set
enrichments, populations of immune infiltrates, stroma and other
clinical variates.

The study is well presented and methodologically sound, although as
the authors have remarked, a direct involment of m6a regulators in
ovarian cancer is far from being proven.

As a minor comment, I would recommend bigger legends and clearer
figures.

Author Response

Thank you very much for your comments and suggestions. A prognostic signature comprising four N6-methyladenosine related long-noncoding RNAs was constructed and verified for OV according to TCGA and GEO database and its predictive performance was confirmed in this article. For your kindness suggestions, we re-edit the legends and figures.

Round 2

Reviewer 1 Report

We agree with the main objective of the authors.We agree with the main objective of the authors.